# Gendered Perspectives on Intimate Partner Violence: A Comparative Study of General Population, Students and Professionals' Beliefs

Iris Almeida [1,2,*], Ana Ramalho [2], Rafaela Morgado [2] and Ricardo Ventura Baúto [1,2]

[1] Egas Moniz Center for Interdisciplinary Research (CiiEM), Egas Moniz School of Health & Science, 2829-511 Caparica, Almada, Portugal; rbauto@egasmoniz.edu.pt
[2] Egas Moniz Forensic and Psychological Sciences Laboratory (LCFPEM), 2829-511 Caparica, Almada, Portugal; aramalho.lcfpem@egasmoniz.edu.pt (A.R.); rafaela.fmorgado@gmail.com (R.M.)
[*] Correspondence: ialmeida@egasmoniz.edu.pt

**Abstract:** Domestic violence is a worldwide crime recognized as a severe violation of Human Rights, which includes Intimate Partner Violence (IPV). The studies remark that the asymmetries in the social relations between men and women result in domination dynamics. Thus, this study analyzed the relationship between gender and IPV beliefs in the general population, university students, and healthcare/safety/justice professionals by comparing IPV legitimization between men and women and with age. The sample was composed by 3413 Portuguese participants, 1551 men (45.4%) and 1826 women (54.6%), aged 18 to 100 ($M = 37.97$; $SD = 18.09$), 1936 participants from the general population (56.7%), 866 university students [e.g., healthcare students] (25.4%) and 611 healthcare/safety/justice professionals [e.g., doctors, psychologists, police officers, lawyers] (17.9%). The sample filled out the Scale of Beliefs about Marital Violence (ECVC), a self-report scale on beliefs about IPV. Results confirmed our hypothesis that men have significantly higher levels of IPV legitimization than women. In accordance with our second hypothesis, significant positive correlations were found between age and IPV beliefs. As age increases, older people tend to be more tolerant of IPV, and young people tend to be less endorsing such IPV beliefs. Finally, we found the hypothesis that university students and healthcare/safety/justice professionals have lower levels of beliefs compared with other participants in the general population. Findings show that we need to work hard with the social evolution in men's and women's beliefs on IPV, reinforcing the importance of targeting IPV prevention by gender and age in the general population but also in students and professionals.

**Keywords:** Intimate Partner Violence; beliefs; gender asymmetries; healthcare/safety/justice professionals; students; general population





## 1. Introduction

Domestic violence is a worldwide crime recognized as a severe violation of Human Rights (Almeida et al. 2023; Soeiro et al. 2023; WHO 2013, 2021a, 2021b), which includes Intimate Partner Violence (IPV). It is defined as any violent behaviors, such as physical, psychological/emotional, sexual, and stalking acts committed by an actual or ex-intimate partner (Neves and Almeida 2020; Almeida et al. 2023; Soeiro et al. 2023). Physical violence included being battered, pushed, slapped, throwing objects at the partner, stabbed, burned, strangled, and even attempted or committed murder (e.g., Du et al. 2021; WHO 2013). Psychological/emotional violence aims to cause harm to the partner's self-esteem through verbal abuse and power/control actions (e.g., Coker et al. 2002; Day et al. 2003; Poehacker et al. 2017). Sexual violence includes coercion and violent acts occurring during intended or further sexual contact (e.g., Black et al. 2011; Pöllänen et al. 2018; WHO 2013). Stalking is related to unwanted communication, following, or threatening that can intentionally create

a sense of fear in victims (e.g., Ferreira et al. 2014). In this sense, IPV is a transversal issue in many socioeconomic, religious, and cultural communities (Baldry 2003; Geffner 2016; Machado et al. 2006; Paulino-Pereira et al. 2017; Treves-Kagan et al. 2019; Tsirigotis and Luczak 2018; WHO 2005).

IPV constitutes one of the most reported types of crime to the authorities. As a result, Governments have approved new structured policies that allow the Criminal Justice System to achieve this problem through the definition of measures for victim protection, offenders' intervention, professional know-how, and the reinforcement of victim support physical structures and networks, in the development of policies that allow facing this sociocultural issue. In the European Union (EU), an inter-country survey was designed to analyze this problem in the 28 Members, finding that the Eastern-European cultures are the ones where IPV is most prevalent (e.g., Lithuania) (Nevala 2017).

Thus, what kind of variables might explain the IPV numbers? One of them that is relevant to study is IPV beliefs that may explain or predispose the violent conduct, understanding the cultural and social framework where they occur to promote the behavioral and mindset change and minimize the impact of this crime (Machado et al. 2006). Some studies (e.g., Neves and Almeida 2020; Machado et al. 2006) intend to search for explanations for these behaviors. The sociocultural issues of society and beliefs (e.g., exposure to violence in early childhood) have a significant influence on the maintenance and perpetuation of these behaviors through the decades being this statement is unanimous among the literature (Godbout et al. 2019; Machado et al. 2006; Pournaghash-Tehrani 2011).

Nonetheless, some studies (e.g., Cinquegrana et al. 2022) underscore that unless there is a fundamental shift in societal attitudes that facilitate, tolerate, and perpetuate IPV, we cannot expect to effectively combat this issue and significantly reduce its alarming prevalence rates. Therefore, if the goal is to curb the incidence of IPV, a primary focus for public education initiatives should be on combatting sexism.

## 2. Sociocultural Issues and Gender

The studies remark that the asymmetries in the social relations between men and women result in domination dynamics (Amâncio and Santos 2021; Guedes et al. 2009). It is accepted that the hegemony of masculinity is observed, suggesting that the exercised domination results in offending toward their partners (Oliveira and Fonseca 2014). On the other hand, this indicates that the feminine role is for obedience and subordination (Gillum et al. 2018; Heise and Kotsadam 2015; Oliveira and Fonseca 2014). Thus, attitudes that approve or excuse IPV are common in sexist societies, which is an understanding that the main authority figure is men, supporting the idea that "men rules" suppress women, and violence has been perceived as "normal" (Cabral and Rodriguez-Díaz 2017; Evcili and Daglar 2020). When analyzing the dynamics between gender and violence, several researchers have shown that power dynamics is a complex phenomenon that can derive from individual, situational, cultural, and social factors (Vieraitis et al. 2008; Neves and Almeida 2020). These studies have shown that men's domination and IPV tolerance and legitimation are a more prominent presence in the male gender due to the existence of patriarchal societies, where the belief that men have a social and cultural ascendant over women favors the development and maintenance of beliefs favorable to IPV (Almeida et al. 2021; Carlson and Worden 2005; Machado et al. 2014).

A meta-analysis examining the correlation between power (defined as control and dominance) and IPV against women, carried out by Ubillos-Landa et al. (2020), established that IPV is fundamentally about control, originating from entrenched patriarchal norms of male dominance within heterosexual relationships. This perspective often results in an evasion of accountability on the part of the offender, as cultural norms legitimize violence and allow men to shift the blame for the violence against the victim. The endorsement of a belief system that affirms men's entitlement to certain privileges within their relationships with women enables the offender to deflect accountability and rationalize the ongoing use of control and dominance (Ubillos-Landa et al. 2020).

### 3. Beliefs, IPV, and Gender

Belief can be defined as a thought, a feeling, or a predisposition to accept that some idea is accurate. Usually, beliefs are considered involuntary, shaped by evidence, independent of context, dependent on what is considered true, and characterized to varying degrees. It could be seen as a tendency to be influenced by one's knowledge about the world in evaluating conclusions and to accept them as true because they are believable and logically valid. Belief bias is often assessed with syllogistic reasoning tasks in which the believability of the conclusion conflicts with logical validity (APA 2020). Violence is a consequence of dysfunctional beliefs, constituted a result of socialization and internalized from an early age, influencing behavior. Thus, it is essential to acknowledge how violence is interpreted by individuals, bearing in mind that this dimension is associated with cultural norms that influence self-perceptions.

The beliefs related to violence influence the perception of people and the considerations we make about the environment. Therefore, this aspect has a significant role in the behavior (Machado et al. 2006) and is crucial to indicate dysfunctional beliefs and thoughts among offenders and the general population (Ferrer-Pérez et al. 2019). Thus, beliefs and attitudes have a base role in the perpetration of IPV (e.g., sexist, patriarchal, and sexually dysfunctional attitudes against partners) (Husnu and Mertan 2017). Nevertheless, victims frequently appear to have beliefs related to the relationship dynamics (e.g., guilty feelings, subservient conduct, and beliefs related to the marriage) (Bosch-Fiol and Ferrer-Pérez 2012; Megías and Montañés 2012; Puente-Martínez et al. 2016). This belief may restrain the victims from seeking help, which constitutes a barrier to the policies that prevent IPV because these behaviors are still culturally legitimized (Almeida et al. 2023; Alves et al. 2019; Barocas et al. 2016; Shen 2014). For example, Mendes and Cláudio (2010) observed that the legitimation of violence mainly results from dysfunctional beliefs that excuse abusive behaviors, which have been internalized since very early influencing behavior (Gonçalves et al. 2021; Silva 2017). According to Machado (2010), the complexity of human relations has a strong contribution to the main conceptions of violence, making it difficult to define and agree about externalization as being violent or not (Ventura et al. 2013). Based on the importance of beliefs and attitudes in abusive relationships, it is essential to identify irrational beliefs and cognitive distortions in criminal and non-forensic populations. Thus, reliable and valid assessment tools are essential for research and intervention purposes (Ferrer-Pérez et al. 2019) because often, research conflates attitudes with social norms or employs attitudes as a proxy for social norms due to the limited availability of valid measures (Shakya et al. 2022).

That is why it is important to study IPV beliefs and attitudes among the general population but also among professionals who could directly or indirectly influence the community against this kind of crime (Table 1).

Regarding gender differences and looking at studies on the general population (e.g., Bucheli and Rossi 2019; Machado et al. 2014; Vandello and Cohen 2008), the results show that men tend to have more beliefs favorable to IPV when compared with women. Additionally, the studies with offenders (e.g., Capaldi et al. 2012; Graham-Kevan 2007), mainly with male samples, revealed accentuated IPV beliefs. Such beliefs can be equally accentuated by age and tend to be a central element in a traditional base belief and in devaluation/trivialization situations that support the protection of family privacy and a consequent lack of responsibility of the offenders for causes external to his will (e.g., Band-Winsterstein and Eisikovits 2010; Bucheli and Rossi 2019; Neves and Almeida 2020). Mookerjee et al. (2021) observed that individuals who hold beliefs rationalizing "wife beating," regardless of gender, tend to increase the likelihood of experiencing IPV. However, it is noteworthy that women endorsing such beliefs face a higher risk of IPV compared with their counterparts who do not, and the impact of a woman's beliefs appears to have a more significant influence than that of men's beliefs in this regard.

**Table 1.** Studies using the Scale of Beliefs about Marital Violence (ECVC) in Portugal.

| Title | Authors | Year | Sample | Results |
|---|---|---|---|---|
| Beliefs about domestic violence: the influence of a training plan on young people (Fernandes et al. 2009) | Maria Isabel Fernandes, Ana Bela Jesus Caetano, Cristiana Salomé Almeida, and Ângela Maria Figueiredo | 2009 | 28 Nursing students between the ages of 20 and 30 | Students' tolerance and/or acceptance of conjugal violence is low; conversely, there is a high acceptance of the "Banalisation of small violence" factor. |
| Professionals' beliefs and attitudes towards conjugal violence. Studies with health professionals, police, and teachers (Machado et al. 2009a, 2009b) | Carla Machado, Marlene Matos, Rosa Saavedra, Olga Cruz, Carla Antunes, Márcia Pereira, Ana Rato, Isa Pereira, Cláudia Carvalho, and Liliana Capitão | 2009 | 226 health professionals, 85 police officers, and 280 teachers | Older and male participants tend to show greater legitimacy of violence. |
| Beliefs about domestic violence in different professional classes linked to the drafting and enforcement of legislation in force (Matos and Cláudio 2010) | Teresa Matos and Victor Cláudio | 2010 | 108 public security police officers, 101 military personnel from the Republican National Guard, 61 prosecutors, 26 deputies, and 12 judges serving in criminal courts. | The participants showed higher values in "legitimation and trivialisation of small violence". The GNR soldiers were the professionals who obtained higher values in all factors and on a broad scale. Men obtained systematically higher values than women. |
| Violence in dating relationships: influence of beliefs and area of training (Machado et al. 2010) | Teresa Sousa Machado, Isabel Maria Macieira, and Maria Conceição Carreiras | 2010 | 100 university students aged between 19 and 39 years old who maintain or have maintained a lasting relationship | Boys are more tolerant of violence; a significant part of the sample reports that they were perpetrators or victims. |
| Beliefs and attitudes of police officers towards violence against women (Coelho 2010) | Alexandra Miranda Coelho | 2010 | 453 public security police | Significant influence of the perceived seriousness and the sense of personal responsibility in the interventions carried out by police officers, occurring only in the face of physical violence associated with repeated forms of violence against women. Men police officers have higher average values for the legitimizing beliefs of violence against women. |
| Violence and intimate relationships in higher education in Portugal: representations and practices (Mendes et al. 2013) | José Manuel Mendes, Madalena Duarte, Pedro Araújo, and Rafaela Lopes | 2013 | 58 university students | Higher education students dissociate themselves from this perception and do not activate behaviors of denunciation or even prevention. |
| Conjugal violence: beliefs of current and future professionals. Involved in response and prevention—law, health, and education (Cabral and Rodriguez-Díaz 2017) | Paula Cristina Cabral and Francisco Javier Rodríguez-Díaz | 2017 | 418 participants aged between 17 and 72 years. | 60% of individuals believe that the phenomenon of conjugal violence has been increasing, 90% believe that there has been a greater sensitivity to the social problem, and 30% verify a considerable tolerance in the population |

**Table 1.** *Cont.*

| Title | Authors | Year | Sample | Results |
|---|---|---|---|---|
| Intervention in domestic violence situations: police attitudes and beliefs (Sani et al. 2018) | Ana Isabel Sani, Alexandra Coelho, and Celina Manita | 2018 | 453 public security police | Higher levels of legitimizing beliefs in conjugal violence are associated with more conditioned police action. |
| Beliefs about conjugal violence and violence in the context of intimate relationships in higher education students (Massano 2018) | Fabiana Carvalho Massano | 2018 | 306 higher education students | Negotiation is the most prevalent conflict resolution strategy among students. The legitimizing beliefs of violence are firmer in students who have already suffered or used abusive conflict resolution strategies in their dating relationships. Men have firmer legitimating beliefs about violence than women. |
| Beliefs scale on marital violence (ECVC): Brazilian version (Moura et al. 2021) | Julliane Quevedo de Moura, Luísa Fernanda Habigzand, Marlene Matos and Mariana Gonçalves | 2021 | 1337 Brazilian adults (general population) | Men with less schooling and with children showed greater agreement with legitimizing IPV beliefs. |

When studying gender differences in university samples, Larsen (2016) found that women tend to be more accurate in identifying this type of situation. Still, in this sample, Larsen (2016) verified that 55% of the participants believed this problem existed on their own University Campus. In Portugal, Neves et al. (2022), in a Portuguese university sample, found that men have more conservative beliefs about gender social relations than women.

Concerning beliefs among health professionals, Briones-Vozmediano et al. (2022) found that the training received on the phenomenon does not seem to be enough because they are not being prepared to deal with a social and emotional component (focused on the biomedical and pathological response resulting from situations of violence). Jack et al. (2021) admitted that this problem resides in pre-graduate training and during career development. Briones-Vozmediano et al. (2022) argue that, although training is not a single condition for the dilution of professionals' beliefs, not least because these derive from their own experiences and personal development, they allow it to function as a barrier or potentiate the reduction of vulnerability arising from their perception of the phenomenon. Martínez-García et al. (2021) concluded that there are still some negative perspectives, referring to a view of the problem as a phenomenon of a particular and personal nature of women that the public health system should not treat. Johnson and May (2015) reinforce that beliefs are one of the most significant barriers to the adequate response of professionals during the care provided. In this line, how the system trains its professionals can help in the way in which the victims themselves are received and forwarded. An example of the usefulness of this training/intervention work with professionals is the study by Arora et al. (2021), which focused on the impact of a program on the knowledge, attitudes, and skills of these professionals on violence, first-line, social, and legal support through referrals. The intervention in this study also included training and changes at the system level to create a supportive ecosystem for professionals in this area, as Sprague and The EDUCATE Investigators (2019) advocates. The results were considered positive, with a direct demonstration of improvement in the performance of professionals in their work services.

At the same time, one of the other professional groups that had contact with IPV is the police, who are responsible for investigating the crime but also, as a result of this expertise, listening to victims and offenders. An optimistic note about these professionals' perspective regarding crime is that Russell (2018) found in his study that the gender of those involved does not present significant differences in the attribution of responsibilities

by the police (i.e., men and women). Women are investigated in the same way, depending on whether they play the victim or suspect in the crime), not attributing tremendous guilt to one over the other, valuing only the facts allegedly committed. However, other studies (e.g., Matos and Cláudio 2010; Ferreira et al. 2022) suggest that male professionals tend to have higher legitimacy in the general context of security forces and criminal justice, leading to a more passive face of the crimes committed.

Many studies relate IPV and beliefs among the general population and professionals. In this sense, this study analyzed the relationship between gender and IPV beliefs in the general population, university students, and healthcare/safety/justice professionals in the Portuguese context.

From this objective, we hypothesize that: (1) Men have a higher level of beliefs about IPV when compared with women (e.g., Graham-Kevan 2007; Capaldi et al. 2012; Machado et al. 2007; Machado et al. 2014; Moura et al. 2021; Vandello and Cohen 2008); (2) the level of IPV legitimization is positively correlated with age (e.g., Band-Winsterstein and Eisikovits 2010; Bucheli and Rossi 2019; Machado et al. 2009a, 2009b; Martinez and Khalil 2017; Moura et al. 2021; Neves and Almeida 2020); (3) healthcare/safety/justice professionals and university students have lower levels of beliefs compared with other participants in the general population (e.g., Ferrer-Pérez et al. 2019).

## 4. Materials and Methods

### 4.1. Participants

The sample was composed by 3413 Portuguese participants, 1551 men (45.4%) and 1826 women (54.6%), aged 18 to 100 ($M$ = 37.97; $SD$ = 18.09), selected by convenience sampling: 1936 participants from the general population (56.7%), 866 university students [e.g., healthcare students] (25.4%), and 611 healthcare/safety/justice professionals [e.g., doctors, psychologists, police officers, lawyers] (17.9%). Regarding educational qualifications, they vary between the 1st Cycle (1st–4th year) and the BSc degree (Table 2).

**Table 2.** Sociodemographic characteristics.

|  | Frequency | Percentage (%) |
|---|---|---|
| Educational qualifications |  |  |
| 1st Cycle (1st–4th year) | 371 | 10.9 |
| 2nd Cycle (5th–6th year) | 350 | 10.3 |
| 3rd Cycle (7th–9th grade) | 296 | 8.7 |
| Secondary education (10th–12th grade) | 758 | 22.2 |
| University students | 861 | 25.2 |
| Bachelor | 28 | 0.8 |
| BSc degree | 749 | 21.9 |
| Marital status |  |  |
| Single | 1706 | 50 |
| Married/Union | 1316 | 38.6 |
| Divorce/Separated | 225 | 6.6 |
| Widow | 166 | 4.9 |

### 4.2. Instrument

Portuguese participants were asked to answer the "Scale of Beliefs about Marital Violence" (ECVC; Machado et al. 2006), a Portuguese self-report scale to assess beliefs about IPV. This instrument is composed of 25 items, scored from 1 to 5 on a Likert scale (*totally disagree* to *totally agree*), grouped into four factors: Factor 1 legitimizing and trivialization of minor violence (e.g., offensive, hitting); Factor 2 legitimization of violence by women's conduct (e.g., infidelity, provocative); Factor 3 legitimization of violence by its attribution to external causes (e.g., unemployment, extra-marital relationships); and Factor 4 legitimization of violence by the preservation of family privacy (e.g., appealing to the concept of privacy and the need to protect families from outside interference). Total scores

can range from 25 to 125 points. The higher the scores obtained on the ECVC, the higher the levels of IPV legitimization. In the current study, Cronbach's alphas were excellent (0.96) for total scores. For the four groups/factors, the Cronbach's alphas can range from excellent to good: Factor 1 (0.96), Factor 2 (0.92), Factor 3 (0.86), and Factor 4 (0.82).

### 4.3. Procedure

Data were collected between 2010 and 2022. Participants were approached in universities and other public or private institutions and surveyed face to face after signing an informed consent. Oral and written informed consent had been obtained. All ethical principles were attended to and conducted by the Declaration of Helsinki, as well as the Code of Ethics of the Order of Portuguese Psychologists and the General Data Protection Regulation. In addition to the above, the present study is included in the One Justice Project: The Forensic Psychology in Justice and Community, approved by the appropriate institution.

### 4.4. Data Analysis

The IBM statistical version SPSS 28 was used to analyze the data obtained. A descriptive statistical analysis was initially performed on the data, followed by a complementary statistical analysis that included Pearson correlations and other tests such as the t-student test and ANOVA. Pearson correlations were performed between the scales and subscales to verify the relationship between variables. Additionally, the following statistical tests were conducted: A t-student test was employed to examine the ECVC scores based on gender. Finally, an analysis of variance (ANOVA) was employed to assess the differences in ECVC scores among professionals, students, and the general population. This test allows for comparing ECVC scores across different participant groups to identify significant variations.

## 5. Results

This study examined the relationship between gender and IPV beliefs in the general population, university students, and healthcare/safety/justice professionals.

To characterize IPV beliefs, Table 3 represents the mean ECVC scores obtained by the total sample. Results show that the total mean score is above the scale middle point, thus showing a prevalence of beliefs that legitimize IPV in the current sample.

**Table 3.** Descriptive statistics for ECVC scores in the total sample (N = 3.413).

| | M (SD) | Range |
|---|---|---|
| Factor 1—legitimizing and trivialization of minor violence | 32.81 (15.42) | 16–80 |
| Factor 2—legitimization of violence by women's conduct | 20.98 (9.44) | 10–50 |
| Factor 3—legitimization of violence by its attribution to external causes | 18.44 (7.09) | 8–40 |
| Factor 4—legitimization of violence by the preservation of family privacy | 13.81 (5.26) | 6–30 |
| Total Factor—IPV legitimization (ECVC) | 86.04 (35.82) | 25–125 |

Comparing the mean ECVC scores between genders (men and women) shows that men have significantly higher levels of IPV legitimization than women (Table 4), where the only factor without a significant difference is the legitimization of violence by the preservation of family privacy (Factor 4). Legitimizing and trivialization of minor violence (Factor 1) were the beliefs with the largest effect size, followed by the legitimization of violence by women's conduct (Factor 2) and legitimization of violence by its attribution to external causes (Factor 3).

**Table 4.** Descriptive statistics and t-student test results for ECVC scores by gender.

| | Men | Women | | |
|---|---|---|---|---|
| | **M (SD)** | | | **Effect Size (Cohen's d)** |
| Factor 1 | 40.08 (16.19) | 26.75 (11.70) | t = 27.85; $p < 0.001$ | 13.918 |
| Factor 2 | 25.36 (9.58) | 17.34 (7.60) | t = 27.27; $p < 0.001$ | 8.558 |
| Factor 3 | 21.63 (6.71) | 15.78 (6.26) | t = 26.31; $p < 0.001$ | 6.466 |
| Factor 4 | 16.15 (4.53) | 11.85 (5.02) | t = 26.06; $p = 0.336$ | 4.800 |
| Total Factor | 103.23 (35.50) | 71.73 (29.17) | t = 28.45; $p < 0.001$ | 32.204 |

Analyzing how IPV level of legitimization varies with age, we found significant positive correlations between age and ECVC scores (Table 5), which means that IPV legitimization increases from younger to older age. Analyzing how the IPV level of legitimization varies with age, we found significant positive correlations between age and ECVC scores.

**Table 5.** Correlation between beliefs and age.

| | Age |
|---|---|
| Factor 1 | 0.209 ** |
| Factor 2 | 0.216 ** |
| Factor 3 | 0.229 ** |
| Factor 4 | 0.282 ** |
| Total Factor | 0.234 ** |

Note: ** $p < 0.01$.

Regarding the general population, university students, and healthcare/safety/justice professionals, the results suggest that there are significant differences in the ECVC scores (Table 6). The general population presents the highest score for IPV legitimization total score in all the factors, followed by professionals and students.

**Table 6.** ANOVA results for ECVC by sample.

| | Professionals | Students | General | | $\eta^2$ |
|---|---|---|---|---|---|
| | **M (SD)** | | | | |
| Factor 1 | 31.13 (15.50) | 29.36 (14.83) | 34.88 (15.81) | F = 43.86; $p < 0.001$ | 0.25 |
| Factor 2 | 20.07 (9.62) | 18.83 (8.97) | 22.23 (9.40) | F = 43.28; $p < 0.001$ | 0.25 |
| Factor 3 | 17.57 (7.05) | 17.05 (6.60) | 19.34 (7.29) | F = 37.45; $p < 0.001$ | 0.21 |
| Factor 4 | 13.17 (5.14) | 12.29 (4.61) | 14.69 (5.38) | F = 70.78; $p < 0.001$ | 0.40 |
| Total Factor | 81.93 (36.11) | 77.54 (33.90) | 91.14 (35.70) | F = 49.43; $p < 0.001$ | 0.28 |

The Welch test rejects the null hypothesis of equal population means (Table 7), which means that ECVC factors differ significantly across samples.

**Table 7.** Robust tests of equality of means.

| | Statistic | df1 | df2 | Sig |
|---|---|---|---|---|
| Factor 1 | 44.691 | 2 | 1446.782 | <0.001 |
| Factor 2 | 44.688 | 2 | 1445.488 | <0.001 |
| Factor 3 | 38.714 | 2 | 1471.183 | <0.001 |
| Factor 4 | 76.549 | 2 | 1496.121 | <0.001 |
| Total Factor | 50.898 | 2 | 1451.246 | <0.001 |

Table 8 shows that there is a significant difference between the samples, confirming that the general population presents the highest score for IPV legitimization total score in all the factors.

**Table 8.** Post hoc Scheffee test.

| | (I) Sample2 | (J) Sample2 | Mean Difference (I-J) | Std. Error | Sig. |
|---|---|---|---|---|---|
| Factor 1 | Professionals | Students | 1.765 | 0.805 | 0.090 |
| | | General | −3.756 * | 0.707 | <0.001 |
| | Students | Professionals | −1.765 | 0.805 | 0.090 |
| | | General | −5.520 * | 0.623 | <0.001 |
| | General | Professionals | 3.756 * | 0.707 | <0.001 |
| | | Students | 5.520 * | 0.623 | <0.001 |
| Factor 2 | Professionals | Students | 1.235 * | 0.493 | 0.043 |
| | | General | −2.164 * | 0.433 | <0.001 |
| | Students | Professionals | −1.235 * | 0.493 | 0.043 |
| | | General | −3.399 * | 0.381 | <0.001 |
| | General | Professionals | 2.164 * | 0.433 | <0.001 |
| | | Students | 3.399 * | 0.381 | <0.001 |
| Factor 3 | Professionals | Students | 0.516 | 0.371 | 0.379 |
| | | General | −1.768 * | 0.326 | <0.001 |
| | Students | Professionals | −0.516 | 0.371 | 0.379 |
| | | General | −2.284 * | 0.287 | <0.001 |
| | General | Professionals | 1.768 * | 0.326 | <0.001 |
| | | Students | 2.284 * | 0.287 | <0.001 |
| Factor 4 | Professionals | Students | 0.875 * | 0.272 | 0.006 |
| | | General | −1.525 * | 0.239 | <0.001 |
| | Students | Professionals | −0.875 * | 0.272 | 0.006 |
| | | General | −2.401 * | 0.211 | <0.001 |
| | General | Professionals | 1.525 * | 0.239 | <0.001 |
| | | Students | 2.401 * | 0.211 | <0.001 |
| Total Factor | Professionals | Students | 4.392 | 1.866 | 0.063 |
| | | General | −9.212 * | 1.639 | <0.001 |
| | Students | Professionals | −4.392 | 1.866 | 0.063 |
| | | General | −13.604 * | 1.444 | <0.001 |
| | General | Professionals | 9.212 * | 1.639 | <0.001 |
| | | Students | 13.604 * | 1.444 | <0.001 |

* $p < 0.001$.

## 6. Discussion

IPV is just part of a seriously widespread crime that relates to domestic violence. A practice firmly repudiated by the international community. It still encounters continuous challenges that go beyond the social policies of the central Governments of each country, but mainly in the resistance to change of a still accentuated fringe of the world population. In Western countries, positioned as socially developed, the lightness and banalization of some of the behaviors analyzed in this study contribute to the perpetuation of this crime that compromises the direct victims but also the entire community and the avoidance of accepting the severity and impacts of IPV.

This study focuses on the most representative element of the challenge for change, the structural belief associated with IPV behavior, and the way in which the individual understands its severity. Without moral scrutiny that disapproves of conduct of this nature, the behavior tends to remain fueled by the idea of impunity that society shares with someone who behaves in this way.

According to the results obtained, it was possible to verify an element that was already anticipated, given the number of studies developed in this area, which highlights the existence of dysfunctional beliefs that perpetuate and lead to justify or trivialize behaviors associated with violence in intimate relationships. These results are similar to those found in national (e.g., Machado et al. 2006) and international (e.g., Ferrer-Pérez et al. 2019) studies.

In a more directed way to the hypotheses present in this investigation, the results obtained confirm Hypothesis 1, "Men have a higher level of beliefs about IPV when compared to women". This result is supported by a wide range of studies that obtained

the same conclusions (e.g., Graham-Kevan 2007; Machado et al. 2014; Capaldi et al. 2012; Vandello and Cohen 2008).

At the same time, when analyzing Hypothesis 2, which referred to "Older people tend to show greater legitimacy of violence", this was also confirmed, being consistent with the international literature that suggests that beliefs favorable to violence in intimate relationships are especially pronounced in older people (Band-Winsterstein and Eisikovits 2010; Bucheli and Rossi 2019; Neves and Almeida 2020). However, it should be noted that none of the analyzed studies (this study included) has a longitudinal research design. Thus, it does not allow us to objectively analyze whether this fact is related to an increase in the level of beliefs based on a multifactorial variable of a temporal nature or whether it is delimited to a historical, contextual, and cultural period that conditioned the maintenance and consolidation of beliefs of this type in older individuals. We believe that this could be a relevant line of research, which will make it possible to understand the impact of measures to prevent and combat the phenomenon of IPV and to identify possible fluctuations in the level of beliefs throughout the life cycle.

Hypothesis 3, and the last one, referred to a central element of our research, where we sought to understand whether the training and performance of professional functions that may have a direct or indirect influence on the reduction (e.g., teachers) and/or combat (e.g., lawyers, police) of the IPV with the community. For this purpose, Hypothesis 3 was formulated, "Professionals and students have lower levels of beliefs compared to other participants in the general population". At this point, framed with the reviewed literature review (e.g., Ferrer-Pérez et al. 2019), we found that the hypothesis is confirmed, with the general population appearing with a higher score, followed by healthcare/safety/justice professionals and, finally, university students. It is in these last two that the analysis can be directed to the need for future clarification. Some studies (e.g., Arora et al. 2021; Johnson and May 2015; Sprague and The EDUCATE Investigators 2019) have already stated that the training of various professionals who directly deal with victims of IPV, such as health professionals, tend to be trained for issues of a clinical and technical-scientific nature, bleaching the elements related to the approach to the victim, psychological first aid or even interviewing techniques appropriate to the context. Thus, given the evidence that professionals have a higher level of beliefs than students, the question is extended to another framework previously discussed: Is it the chronological age of the participants that negatively influences their perspective on the IPV or the desensitization towards the theme, given the gap between basic training and field practice? Could other variables associated with the performance of the profession accentuate these beliefs? We understand that the extended analysis of these variables, consolidating a more concrete identification of theparticipant's level of social and psychological training and the development of relational and interpersonal skills, could accentuate the levels of awareness of professionals and students, potentially having positive impacts on the general population, given the positions these professionals occupy in society. Thus, other factors (e.g., exposure to violence in early childhood) can have a significant influence on beliefs and attitudes (Godbout et al. 2019; Machado et al. 2006; Pournaghash-Tehrani 2011). Despite the results obtained, we are aware that this investigation has some limitations, namely the use of a self-reporting tool because people feel vulnerable to provide personal information and tend to respond according to social desirability. The sampling method did not assure the representativity of the study sample as a whole, nor all groups that work directly or indirectly with IPV, and cross-cultural studies are needed for a better understanding of IPV beliefs. Another limitation is the age range as well as the lack of other explanatory variables (e.g., personality, exposure to violence, cognitive factors). In conclusion, it is important to mention that although there are no clear indications of the COVID-19 pandemic period affecting the participants' perspective on IPV, future result analyses should exercise added caution, in line with any empirical knowledge that may emerge, to minimize the potential for bias. Despite these limitations, the results do provide important contributions to the study of IPV beliefs.

### 7. Conclusions

Domestic violence is a phenomenon that occurs in different spheres of society. Violence-related issues focus on the differentiation of gender roles in a society where patriarchy is especially present. Although legal and social advances have been witnessed, the differentiation of opportunities and how men and women behave continue to be seen and accepted, or not, equally influencing career choices, career access and development, and influence on the family nucleus. Also, in accessing and selecting professions and their training, there are differences in how men and women see and treat the profession and how they approach and relate to the crime of domestic violence and, consequently, IPV. The contact of some professionals with this reality is also anchored in personal beliefs and their evolution and maintenance throughout their professional lives. Given the results obtained in our study, which are parallel to others obtained internationally and nationally, it is urgent to define action plans in the training curricula of professionals. These programs should not be limited to interventions focused on the professional's technical skills; they should also reinforce the soft skills training that could be central factors in providing better quality services to direct and indirect victims. Intervention in the psychosocial dimension emerges as an act of collective citizenship, which cannot start only from political initiative but mainly from an individual will. This should be promoted from pre-school to adulthood, which may mean adapting training curricula in a college context in the case of more qualified professionals. Results show that we need to work hard with social evolution in men's and women's beliefs about IPV, shedding light on how women may be particularly vulnerable to victimization and men to offending, thus reinforcing the importance of targeting IPV prevention by gender. Greater awareness may not be enough to counteract the rise in IPV statistics, but it works in favor of an increase in reporting, gradually giving voice to a once-silent crime. Despite the results of this study, we think it is important to look at the future and study other important variables that can explain beliefs and attitudes in age, profession, and gendered perspectives, such as cognitive factors, personality, attachment, and other social issues. Given the evolution of community intervention programs, it seems appropriate to consider understanding their impact on beliefs that legitimize IPV. Therefore, a longitudinal study (with specific milestones) related to the training of healthcare/safety/justice professionals may provide a more suitable perspective on the impacts of current public policies.

**Author Contributions:** Conceptualization, I.A., A.R., R.M. and R.V.B.; methodology, I.A., A.R., R.M. and R.V.B.; formal analysis, I.A., A.R., R.M. and R.V.B.; investigation, I.A. and R.V.B.; resources, I.A. and R.V.B.; data curation, I.A., A.R., R.M. and R.V.B.; writing—original draft preparation, I.A.; writing—review and editing, I.A.; supervision, I.A.; project administration, I.A.; funding acquisition, I.A. All authors have read and agreed to the published version of the manuscript.

**Funding:** The authors thank the FCT/MCTES—Foundation for Science and Technology, and I.P. for the financial support to CiiEM (UIDB/04585/2020) through national funds.

**Institutional Review Board Statement:** All ethical principles were attended to and conducted by the Declaration of Helsinki, as well as the Code of Ethics of the Order of Portuguese Psychologists and the General Data Protection Regulation. In addition to the above, the present study is included in the One Justice Project: The Forensic Psychology in Justice and Community, approved by the appropriate institution (Process nº 1231).

**Informed Consent Statement:** Written informed consent was obtained from all participantes involved in the study.

**Data Availability Statement:** The data presented in this study is available upon formal request made to the corresponding author. It remains confidential and is not publicly accessible due to its sensitive nature.

**Conflicts of Interest:** The authors declare no conflict of interest with respect to the research, authorship, and/or publication of this article.

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
