# Peer review of "Gendered Perspectives on Intimate Partner Violence: A Comparative Study of General Population, Students and Professionals’ Beliefs"

_socsci, doi:10.3390/socsci12090528_

Round 1
Reviewer 1 Report
The authors have examined a topic that is important and significant for science and practice in a large sample. The hypotheses are derived from the theoretical-empirical background in a comprehensible manner.
In chapter 2 under 2.2 all Cronbach's alpha coefficients of the scales and for the total value should be presented. It would also be good to know whether the individual scales/factors for the sample could be replicated in a factor analysis.
Under chap. 3, the Levene test for normal distribution should be listed and presented for the ANOVAs. A dedicated statistical analysis for control variables should also be carried out (including checking the influence of marital status, education and being widow (high number ==> why?).
The power/effect size should be added to Tables 4 and 5. The same would be true for the Table 6; maybe a post-hoc Scheffee test would be good). Accordingly, the three hypotheses are treated and answered in a rather unfounded manner and vaguely through insufficient statistical analyses. In addition, a regression analysis could be carried out to clarify the largest effect on the overall factor ECVC.
Unfortunately, both the introduction and the discussion leave out other possible factors influencing the perception of IPV. Possible influences here are violence experienced in childhood, intelligence, personality factors (e.g. low agreeableness; agreeableness increases, for example, especially in men, so that the age effect can also be attributed to a personality change, low empathy) or other social influencing factors. The authors should supplement this and at least touch it on in the discussion.
The manuscript should be revised and improved based on the aspects mentioned. Overall, this is a very important and good paper. It just needs to be reworked a bit. From my point of view, these are feasible, but also important points.
good luck with the revision or addition. I look forward to the finished paper
Author Response
Dear Reviewer,
We send you the revisions suggest about original research article entitled “Gendered Perspectives on Intimate Partner Violence: A Comparative Study of Professionals and Students' Beliefs”, for publication in Social Sciences – New Directions in Gender Research -2nd edition.
Thank you for your consideration of this manuscript and for your time and consideration!
We look forward to your response.
Sincerely,

Reviewer 2 Report
Review
The manuscript is adapted to the topics of the journal. It is an interesting job within the field of abuse.
Some issues are pointed out that the authors should improve.
Introduction:
In the section on sociocultural issues and gender, beliefs are discussed. However, there is a section on beliefs. I think it would be better in that section.
Materials and method:
The age of the participant ranges from 18 to 100. It is a very wide interval. Could different groups be made based on age? The variables analyzed could change.
Results:
The effect sizes are missing from the table.
It could be interesting to analyze the relationship between marital status and beliefs. In addition, the age variable should be controlled. Do different intervals.
Discussion:
The authors can improve the discussion. They should support their results with the references used in the introduction. Increase the number of references.
References:
Review. There are some errors such as: lack of page, journal without italics, ….
Author Response

(The authors gave the same response as above.)

Reviewer 3 Report
Thank you very much for allowing me to read your text. I find it an interesting and valuable topic. I think that you have done a good job and that theoretically it is a text that shows a robust previous research.
Even so, I consider that there are limitations that make the article should be reworked.
1. The introduction is weak. It seems to be part of the theoretical framework. In the introduction I expect to see the objective of the article, its structure and how it will achieve what is proposed.
2. The keywords need to be improved. These should be truly identifying of the study and not random words.
3. There are two concerns that I ask for special attention. The first one is associated with the fact that the study is not geolocalized, that is, it does not explain where it was done. Why is this relevant? Because this issue is strongly associated with cultural imaginaries. Unless the authors claim that the population survey is from all over the world, which I do not believe because of the number, they will need to explain where they did it. This study is not the same in Latin America, Asia or Denmark. Once they clarify this, it will be necessary for the theoretical framework to explain the cultural characteristics of the region and how this could influence the topic in general.
4. The other point of concern is that the survey was done over more than a decade. In matters of beliefs, the samples have two senses... if they are done in a single moment we can appreciate how the belief impacts on a subject, which is what this text does... but if the data are too long, it is necessary to consider in the explanation the way in which the belief has evolved. In this sense, a decade is too long and can influence the changes in beliefs in the results... especially in the pandemic years when violence in homes detonated sharply. In this case, it is necessary to explain this methodologically, to divide the applications by periods, to show if there are differences or evolution and of course, to include a complete section explaining possible different situations in the pandemic years.
5. The conclusions are weak. It needs to include the limitations of the study, which I see several, as well as possible lines of future research.
6. The bibliography is not in the format requested by the journal and I would suggest including more texts from the last 4 years.
I think it is a good study, but it needs to work better methodologically.
Author Response

(The authors gave the same response as above.)

Round 2
Reviewer 3 Report
Many thanks to the authors for the considerable improvement of the text. It is important to note that these comments are not meant to annoy you, but to make the article clearer for any reader. Articles should be made with the readers in mind.
Although a significant change has been made, I would appreciate:
1. It needs to be clearer that it is a Portuguese community from the beginning. It is necessary to consider that the characteristics of Portugal can influence the results, since after all it is a subject that depends a lot on local imaginaries. Please make it clear from the abstract itself that it is a group of Portuguese students, as well as in the introduction. It is not enough to mention it in the methodology.
2. I am still concerned about the lack of consideration of COVID19. I understand the argument of your response, however, the pandemic went beyond an isolated situation that can be overlooked. It changed many paradigms, including those associated with gender issues. In this regard, I suggest that if you are not going to consider it, at least mention it within the study limitations, so that the reader knows that this bias in the results exists. Since they do not consider it, the best thing to do is to make it clear.
These are two details that I would be grateful if you would pay attention to
Author Response
Dear Editor,
We appreciate your recommendations, as well as those from reviewer 3. Below, you will find our responses and references to the changes made.
Thank you
